# FADS1 and FADS2 Gene Polymorphisms Modulate the Relationship of Omega-3 and Omega-6 Fatty Acid Plasma Concentrations in Gestational Weight Gain: A NISAMI Cohort Study

**DOI:** 10.3390/nu14051056

**Published:** 2022-03-02

**Authors:** Jerusa da Mota Santana, Marcos Pereira, Gisele Queiroz Carvalho, Maria do Carmo Gouveia Peluzio, Iúri Drumond Louro, Djanilson Barbosa dos Santos, Ana Marlucia Oliveira

**Affiliations:** 1Center of Health Sciences, Universidade Federal do Recôncavo da Bahia, Santo Antonio de Jesus, Avenida Carlos Amaral, R. do Cajueiro, 1015, Cruz das Almas 44574-490, Bahia, Brazil; djanilsonb@gmail.com; 2Collective Health Institute, Universidade Federal da Bahia, Rua Basílio da Gama, Salvador 40110-040, Bahia, Brazil; anamarluciaoliveira@gmail.com; 3Department of Nutrition, Universidade Federal de Juiz de Fora-Campus Avançado de Governador Valadares, Governador Valadares 35010-17, Minas Gerais, Brazil; giseleqc@outlook.com; 4Nutrition and Health Departament, Universidade Federal Viçosa, Viçosa 36570-900, Minas Gerais, Brazil; mpeluzio@ufv.br; 5Center for Human and Molecular Genetics, Universidade Federal do Espírito Santo, Vitória 29500-000, Espírito Santo, Brazil; iurilouro@yahoo.com

**Keywords:** fatty acid desaturases, weight gain, pregnancy, polymorphism, genetic

## Abstract

The polymorphisms of fatty acid desaturase genes FADS1 and FADS2 have been associated with an increase in weight gain. We investigated FADS1 and FADS2 gene polymorphisms and the relation between ω-3 and ω-6 fatty acid plasma concentrations and gestational weight gain. A prospective cohort study of 199 pregnant women was followed in Santo Antônio de Jesus, Brazil. Plasma levels of polyunsaturated fatty acids (PUFAs) were measured at baseline and gestational weight gain during the first, second, and third trimesters. Fatty acid recognition was carried out with the aid of gas chromatography. Single nucleotide polymorphisms (SNPs) were genotyped using real-time PCR. Statistical analyses included Structural Equation Modelling. A direct effect of FADS1 and FADS2 gene polymorphisms on gestational weight was observed; however, only the SNP rs174575 (FADS2) showed a significant positive direct effect on weight over the course of the pregnancy (0.106; *p* = 0.016). In terms of the influence of SNPs on plasma levels of PUFAs, it was found that SNP rs174561 (FADS1) and SNP rs174575 (FADS2) showed direct adverse effects on plasma concentrations of ω-3 (eicosapentaenoic acid and alpha-linoleic acid), and only SNP rs174575 had positive direct effects on plasma levels of ARA and the ARA/LA (arachidonic acid/linoleic acid) ratio, ω-6 products, while the SNP rs3834458 (FADS2) had an adverse effect on plasma concentrations of EPA, leading to its increase. Pregnant women who were heterozygous and homozygous for the minor allele of the SNP rs3834458 (FADS2), on the other hand, showed larger concentrations of series ω-3 substrates, which indicates a protective factor for women’s health.

## 1. Introduction

Obesity is a public health issue affecting women from all continents and of different ethnic origins. Excessive weight gain (GWG) in pregnancy may be related to postpartum weight retention (PPWR) [1], associated with the development of obesity in the immediately subsequent cycles of a woman’s life [2].

Observational studies have identified that pre-pregnancy obesity and excessive GWG have been associated with adverse maternal and child outcomes [3], increasing the risk of gestational diabetes, pre-eclampsia, large-for-gestational-age births, excessive postpartum weight retention, and offspring obesity [4].

Pregnancy is marked by higher caloric and nutritional needs, which, if not met, may favor greater maternal weight gain. In addition, genetic factors may contribute to trigger obesity in pregnancy [5,6].

This relationship can be understood through the theory of nutrigenetics, which points to the effects of biological markers, such as genetic polymorphisms, in the individual’s response to food [7,8]. Based on this theory and on the scientific information that indicates a relationship between nutritional and genetic factors associated with overweight in the gestational period, the single nucleotide polymorphisms (SNPs) of the FADS genes (desaturase genes) stand out.

The FADS2 and FADS1 genes are responsible for coding the desaturase enzymes (delta-5 and delta-6, respectively), which are crucial in the endogenous conversion process of 18-carbon PUFAs (polyunsaturated fatty acids) into very long-chain fatty acids, such as arachidonic acid (ARA), docosahexaenoic acid (DHA), and eicosapentaenoic acid (EPA) [9]. SNPs in these genes are associated with the bioavailability of PUFAs (ω-3 and ω-6) [10,11,12].

Thus, scientific evidence reveals that SNPs in the FADS1 and FADS2 genes can reduce the function of the desaturase enzymes and influence the bioavailability of PUFAs ω-3 and ω-6 in various tissues, such as plasma, breast milk, and red blood cells [13,14], favoring the plasma increase of LA and ALA and a reduction of ARA, EPA, and DHA [10,11]. Plasma excess of LA may be related to an increase in body weight [12] and to the inflammatory process [15,16]. The relationships between FADS1 and FADS2 gene polymorphism, fatty acid bioavailability, and infant outcomes are well elucidated; however, the influence on maternal weight gain remains unelucidated among socioeconomically vulnerable pregnant women living in low- and middle-income countries such as Brazil.

From this perspective, this study aims to evaluate the influence of SNPs of the FADS1 and FADS2 genes expressed in the plasma concentrations of PUFAs (ω-3 and ω-6) and the relationship with weight gain in pregnancy.

## 2. Materials and Methods

### 2.1. Study Design and Population

This is a cohort study involving pregnant women in the Family Health Units of the municipality of Santo Antônio de Jesus. The study included pregnant women seen at fourteen Family Health Units located in the municipality’s urban area, between August 2013 and December 2014 were invited to participate; rural units were not accessed due to the difficulty of transportation to the municipality’s urban area. These women were participants in the NISAMI (Nucleus of Research on Maternal and Child Health) Cohort.

### 2.2. Sample Size

Data was complete for 185 women of the 250 who were followed up, registering a 26% loss, as presented in the cohort flowchart (Figure 1). The sample power was calculated based on the prevalence of 48.1% overweight during gestation identified in pregnant women from the same municipality where this study was developed [17]. Under these conditions, the sample of 185 individuals has a 90% power (1-b) of detecting an association between a FADS gene polymorphism and gestational weight gain. The flowchart of the cohort is shown in Figure 1.

### 2.3. Exclusion and Inclusion Criteria

The pregnant women were clinically healthy, aged 18 years or older, living in the urban area of the municipality of Santo Antônio de Jesus, with a gestational age of 34 weeks or less at baseline and receiving prenatal care in the Public Health System.

Exclusion criteria adopted in the study included multiple pregnancy; adherence to a vegan diet; kidney, contagious, immune, and metabolic diseases; and HIV positivity. The absence of ultrasonographic confirmation of gestational age is also part of the exclusion criteria. However, these conditions were not identified among the selected pregnant women.

### 2.4. Data Collection

The data collection was performed after a pilot study to calibrate the instrument. At the baseline of this study, when a pregnant woman was included in the study, which occurred in the Family Health Units where they received prenatal care, pregnant women signed a written, informed consent form and answered the questionnaire with socioeconomic, demographic, health, and obstetric information. An appointment was then scheduled for blood collection in the clinical analysis laboratory in the city.

### 2.5. Plasma Fatty Acid Composition

The pregnant women were referred to a clinical laboratory in the city, where their anthropometry was evaluated. The blood collection was carried out at least 12 h after overnight fasting. On the day before the blood collection, the pregnant woman received instructions on the exam protocol via telephone. The instructions included a 12 h fast in addition to abstention from physical activity 24 h prior to the examination and from alcohol 72 h prior to the examination. Eight milliliters of venous blood was collected using a tube containing EDTA.

Plasma was separated from other blood components by centrifugation at 2500 rpm for 15 min. The plasma used for extracting the plasma profile of fatty acids was frozen immediately after centrifugation in liquid nitrogen and sent to the Laboratory of Nutritional Biochemistry of the Federal University of Viçosa (UFV), where the plasma concentrations of polyunsaturated fatty acids were performed.

For lipid extraction, plasma samples were thawed at room temperature, and lipids were extracted from plasma by the Folch method (Folch solution: a 2:1 chloroform/methanol solution) [18]. Briefly, 1.9 mL of Folch’s solution (chloroform:methanol, 2:1) was added to the 1 mL plasma sample and vortexed for 2 min. Then, 0.4 mL of pure methanol was added to this solution, and it was centrifuged for 10 min at 3000 rpm. The supernatant was transferred to a clean and dry tube, and 0.8 mL of chloroform and 0.64 mL of NaCl solution (0.73%) were added, vortexed for 30 s and centrifuged for 10 min at 3000 rpm. The upper phase was discarded, and the inner wall of the tube was washed with 1 mL of Folch’s solution. The upper phase was discarded again, and the tube was left in a culture oven with cut-off oxygen circulation, at 37 °C overnight, until dry.

The derivatization step was performed using the Hartman and Lago method (1973) [19]. Upon removing the dry tubes from the oven, 1.2 mL of saponification reagent (NaOH in methanol) was added and heated in a water bath at 80 °C for 20 min, with the tubes screwed closed. Then, 6.0 mL of esterification reagent was added and heated in a water bath at 80 °C for 20 min, with the tubes closed. After this step, the tubes remained at room temperature for approximately 5 min, until reaching a temperature close to 40 °C. Then, 0.5 mL of hexane and 3.0 mL of NaCl solution (20%) were added and homogenized by vortexing. The upper phase was transferred to a labeled amber vial. The tube walls were washed with 0.5 mL of hexane, and the upper phase was transferred to the amber flask. The solvent was dried with nitrogen, and the sample stored in a freezer at −20 °C awaiting chromatographic analysis.

The fatty acid methyl esters were identified by means of gas chromatography using the Shimadzu^®^ gas chromatograph with a Carbowax capillary column (30 m × 0.25 mm) SP-2560 (biscianopropil polysiloxane), 100 m and 0.25 mm diameter, and a flame ionization detector (FID). The programming of the analysis presented an initial temperature of 140 °C, being isothermic for 5 min, and a posterior heating of 4 °C per minute up to 240 °C, maintaining this temperature for 30 min. The temperature of the vaporizer was 250 °C, and the temperature of the detector was 260 °C. The carrier gas used was nitrogen at 20 cm/second, at 175 °C. The split of the sample in the injector was 1/50, and 1 μL of the solution was injected.

In order to record and analyze the chromatograms, the device was coupled to a microcomputer using the GC Solution program. The compounds were separated, and the fatty acids were identified by comparing the retention times of the sample esters with the F.A.M.E. mix reference standard (Sigma-Aldrich^®^, Bellefonte, PA, USA).

Fatty acid relative content was calculated by the GC solution program and was based on the contribution of each fatty acid to the total area of the chromatogram, expressed as fatty acid percentage (%).

### 2.6. Genetic Analyses

For genetic analysis, the genomic DNA was extracted within 72 h of collection from the peripheral blood using the Qiagen FlexiGene^®^ DNA Kit (Qiagen, CA, USA). The DNA was stored at −20 °C until analysis.

SNP selection was made according to their relation with plasma concentrations of long-chain polyunsaturated fatty acids in children and pregnant women [20]. SNPs selected for analysis were rs174575 and rs174561, located in the intronic region of the FADS2 and FADS1 genes, respectively, and the rs3834458 polymorphism, located in the intergenic region between the FADS1 and FADS2 genes, in the promoter region of the FADS2 gene (Appendix A).

Genomic DNA (deoxyribonucleic acid) was extracted. SNP genotyping was performed through real-time Polymerase Chain Reaction (PCR) using TaqMan^®^ assays (Applied Biosystems, Foster City, CA, USA). Detailed information on the genotyping is reported elsewhere [20].

### 2.7. Anthropometric Measurements

In the study follow-up between September 2013 and December 2015, gestational weight gain was monitored. The anthropometric measurements (weight and height) were carried out in the 1st, 2nd, and 3rd trimesters.

The pregnant woman’s weight and height at baseline were measured in duplicate and recorded on an individualized form. The other measurements were performed by Family Health Unit professionals duly trained by the project team, and weight information was collected from the pregnant woman’s chart or card. To measure maternal weight, a Filizola^®^ (Filizola, São Paulo, Brazil) model mechanical scale with a 150 kg capacity and a sensitivity of 100 g was used.

For the measurement of weight, the pregnant woman remained barefoot, with an empty bladder and wearing light clothes, positioned on the scale’s platform with her body weight equally distributed between her feet [21]. For height, a Sanny^®^ stadiometer (São Paulo, SP, Brazil) with a 2000 cm capacity and a sensitivity of 0.1 cm was used. The pregnant woman was measured barefoot, without head accessories, positioned vertically with arms extended along her body, heels together, and head positioned on the Frankfurt plane. Heels, buttocks, shoulder blades, and head came in contact with the vertical surface of the instrument [21]. A maximum variation of 0.5 cm for length measurement and 100 g for weight was accepted [21].

Measurements of weight during pregnancy were performed by appropriately trained prenatal services staff at the Family Health Units and recorded on the pregnant woman’s medical chart and card. Pre-gestational weight was also collected from the pregnant woman’s card.

Body Mass Index (BMI) was used to assess the pregnant woman’s anthropometric status. BMI was obtained by the formula weight/height^2^, using the means of the weight and height values, and evaluated according to the curve of Atalah et al. (1997) [22]. Pre-gestational BMI was classified according to WHO parameters [23].

To calculate gestational weight gain, we considered the difference between weight in the third trimester and pre-gestational weight. The classification was performed based on IOM recommendations [23]. Thus, if a woman starts her pregnancy with a low weight, it is expected that she will gain from 12.5 kg to 18 kg; for eutrophic pregnant women, a gain of 11.5 kg to 16 kg is expected; in the case of overweight, the weight gain range is 7 kg to 11.5 kg; and if the pregnant woman is obese, the expected gain is 5 kg to 9 kg. Weight gain outside the upper and lower limits of these parameters classifies, respectively, low weight gain and high weight gain in the gestational period.

Gestational age was calculated using the gestogram technique, which consists of marking the first day and month of the last menstruation and the day and month of the current consultation on a calendar and making a note of the number of the gestational weeks, which was later confirmed by ultrasound.

### 2.8. Statistical Analysis

In this study, gestational weight gain is the dependent variable and plasma concentrations of polyunsaturated fatty acids (PUFAs) and the SNPs rs174575 (FADS2), rs174561 (FADS1), and rs3834458 (FADS2) are the primary exposure variables. Each SNP was categorized according to genotype: homozygous for the major allele (MM), heterozygous (Mm), and homozygous for the minor allele (mm). Risks were considered to be (1) Heterozygous-Mm and homozygous for the minor allele-mm and protection (0) Homozygous for the major allele (MM).

In the descriptive statistical analysis, mean and standard deviation were adopted for the continuous variables and prevalence for the categorized variables, in order to describe the study sample. Asymmetric variables were evaluated by the Kolmogorov-Smirnov test or by means of scatter plots and histograms. Normal distribution was observed.

Bivariate analysis was adopted in order to identify the statistical significance of the association between independent variables and outcome. Variables whose associations had a value of *p* ≤ 0.20; were selected for inclusion in the multiple analysis. The variable DHA had a value of *p* > 0.2; due to theoretical strength it was included in the multivariate analysis, but without statistical association.

In order to evaluate the association between plasma concentrations of ω-3 and ω-6, the FADS1 and FADS2 gene polymorphisms, and gestational weight gain, we adopted Structural Equation Modeling (SEM), a multivariate linear model that evaluates the direct and indirect relationships of exogenous variables (predictors), endogenous variables (response variable), and intermediate variables (covariates).

In order to use SEM, it is essential to consider a consistent theoretical assumption in regard to the probable causal relationships that may exist in the set of variables of the study’s database [24].

Among the observable variables, emphasis was placed on maternal education (in years), maternal age, gestational week, gestational alcohol consumption, pre-gestational anthropometric status, FADS1 and FADS2 gene polymorphism, and ω-3 and ω-6 plasma concentrations.

Subsequently, a series of equations were estimated simultaneously, in which the relationship between the same variables could be interchangeable. Thus, a direct effect between ω-3 and ω-6 plasma concentrations, FADS1 and 2 gene polymorphism, and gestational weight gain was tested. The indirect effect of SNPs on maternal weight gain was also investigated.

In order to assess the model’s goodness of fit, the root mean square error of approximation (RMSEA) was used. To evaluate the direct and indirect effects of the studied relationships, standardized coefficients were adopted and interpreted according to the criteria proposed by Kline (2011) [25]: small-effect CP (0.10 and −0.10), medium-effect CP (0.30 and −0.30) and strong-effect CP (>0.50 and >−0.50).

## 3. Results

### 3.1. Description of Participants

All the data were available for 185 of the 250 women fit to participate in the study. However, when comparing weight gain according to age, maternal education, and gestational week among the sample losses (*n* = 65) and the women who remained in the study, no statistically significant differences were detected in the selected associations (data not shown).

The sample was composed of women with a mean age of 26.74 years (SD: 5.96), with an average weight gain throughout pregnancy of 12.84 kg (SD: 4.95); 37.3% of pregnant women showed excessive weight gain. Table 1 and Appendix A presents the sociodemographic, obstetric, nutritional, and genetic characteristics of the pregnant women. The highest weight gain was observed in older pregnant women with the highest pre-gestational BMI and who lived with the largest number of people at home.

Regardless of the pregnant women’s genetic characteristics, the distribution of genotypes was consistent with the Hardy-Weinberg equilibrium, which indicates that allelic frequencies will remain constant across generations. The genotypic frequency of the FADS1 (rs174561) and FADS2 (rs174575 and rs3834458) genes is presented in Appendix A. It was found that the frequency of the minor allele of the polymorphisms under investigation ranged from 22.7% to 32.2% among pregnant women (Appendix A).

### 3.2. Main Analysis

Table 2 describes the standardized coefficients of the association between FADS gene polymorphisms, consumption patterns, PUFAs plasma concentrations, and influence on a woman’s weight during pregnancy.

It was found that the direct effect of the FADS1 and FADS2 gene polymorphisms on gestational weight, adjusted for pre-gestational BMI, maternal age, years of schooling, maternal alcoholism, and gestational week, varied significantly; however, only the rs174575 (FADS2) SNP showed a positive and significant direct effect on weight throughout pregnancy (0.11), indicating that pregnant women homozygous for the minor allele and heterozygotous for this SNP have a higher risk of gestational weight gain (Table 2). The rs174575 (FADS2), rs174561 (FADS1), and rs3834458 (FADS2) variants had direct effects on PUFAs plasma concentrations, with different results according to the type of fatty acid (Table 2).

SNP rs174561 (FADS1) revealed a direct negative effect on plasma levels of ALA (α-linolenic acid) (−0.25) and EPA (eicosapentaenoic acid) (−0.25). The SNP rs174575 (FADS2) also had a direct negative effect on the plasma proportions of EPA (−0.12) and on the EPA/ALA (eicosapentaenoic acid/alpha-linolenic acid) ratio (−0.12), and a positive direct effect on plasma concentrations of ARA (arachidonic acid) (0.24) and the ARA/LA (arachidonic acid/linoleic acid) ratio (0.21), which shows that pregnant women homozygous for the minor allele and those heterozygous for these polymorphisms in the FADS1 genes and FADS2 showed lower plasma concentrations of ω-3 PUFAs, and those homozygous for the minor allele and heterozygous for the rs174575 polymorphism showed an increase in the metabolism products of ω-6 PUFAs. Meanwhile, the SNP rs3834458 (FADS2) had an inverse effect on the plasma concentrations of EPA, promoting their increase (0.27) (Table 2).

The SNP rs174575 (FADS1) was also found to have a positive direct effect on ARA plasma fractions, the ARA/LA ratio, and pre-pregnancy BMI, indicating that women homozygous for the minor allele and heterozygous for this variant have higher plasma concentrations of PUFA ω-6 products, which has an impact on pre-pregnancy BMI increase (Table 2).

The total effect (sum of direct and indirect effects) of SNPs on gestational weight showed a pattern similar to that identified in the direct effect. Only the SNP rs174575 (FADS2) had a positive and significant effect on weight gain during pregnancy. It was observed that the most significant effect of SNPs on gestational weight was direct, with a small amount of indirect effect (Table 2).

Maternal age (0.28) and gestational age (0.12) were variables that had positive direct effects on gestational weight. Thus, women who had the highest increase in gestational weight were those who were older, with a more advanced gestational week and who were alcoholics.

The result of the evaluation of the SNPs’ indirect effect on gestational weight mediated by PUFA plasma concentrations did not indicate a significant effect (Table 3).

## 4. Discussion

The results of this investigation reveal that pregnant women who bear the minor allele of the rs174575 polymorphism of the FADS2 gene had greater gestational weight gain and also show that this polymorphism was associated with higher plasma concentrations of ω-6 PUFAs (ARA and ARA/LA) and lower concentrations of ω-3 PUFAs (EPA; EPA/ALA; ALA). Given these results, it is possible to associate the rs174575 polymorphism of the FADS2 gene with an increase in gestational weight and the elevation of PUFA plasma concentrations of the ω-6 series and lower plasma concentrations of PUFA ω-3. It can be assumed that this FADS2 polymorphism induces lower availability of ω-3 and higher availability of ω-6 acids. Higher availability of ω-6 and lower availability of ω-3 increase the accumulation of adipose tissue and, consequently, increase the risk of gestational weight gain.

However, SNP rs3834458 (FADS2) had an inverse effect on EPA concentrations, indicating that it can increase plasma concentrations of ω-3 substrates and thus be protective for women’s health, avoiding excessive weight gain, by adjusting this parameter during pregnancy.

The scientific evidence available to date confirms that the ω-3 (EPA and DHA) PUFAs present as potential reducers of body fat deposits by acting on the modulation of lipid metabolism, promoting lipolysis and fatty acid oxidation and inhibiting lipogenesis [26].

The results of this study can be interpreted through the lens of nutrigenomics, and they may indicate that endogenous conversion of PUFA is also influenced by the individual’s genetic profile [13,27], and therefore the presence of FADS1 and 2 SNPs may modulate the plasma concentration of ω-3 and ω-6 [9,10,11,12,28].

The results of the present study are in agreement with this theory, and other empirical investigations have also reported similar results, revealing that individuals homozygous for the minor allele of the rs174575 (FADS2) and rs174561 (FADS1) SNPs exhibit lower activity of the desaturase enzymes, with a plasma increase of LA and ALA and a reduction of ARA, EPA, and DHA [10,12,13], which may provide higher levels of arachidonic acid and inflammatory disorders [15,16] due to increased LA and reduced EPA and DHA, as well as increased body weight and/or obesity [12,29].

Regarding the frequency of the homozygous genotype for the minor allele of the evaluated SNP-type polymorphisms of the FADS1 (rs174561) and FADS2 (rs174575) genes, occurrence ranged from 5.6% to 7.2% in the population under study, respectively. These values are lower than those recorded in the scientific literature for the general population (11% and 40%) [13]. It is worthy to note that these values include male and female populations and differ from the population of this study, which is only female.

Epidemiological investigations into the impact of FADS gene polymorphisms on maternal and child health are advancing. Although evidence about this relationship is still being gathered, current knowledge reveals that the minor alleles of FADS1 (rs174561) and FADS2 (rs174575) gene variants alter the bioavailability of LCPUFAs in plasma and breast milk, also expressed as lower DHA and EPA concentrations in both gestational plasma and breast milk [11,13,16].

An intervention study on a cohort of pregnant women revealed that the minor alleles of the FADS1 (rs174533) and FADS2 (rs174575) gene polymorphisms were associated with reduced EPA and DHA plasma concentrations, red blood cells, and phospholipids. However, supplementation with 600 mg per day of DHA during the last two trimesters of pregnancy significantly increased the status of this nutrient in maternal plasma [30].

This physiological response in DHA plasma concentrations after supplementation reflects the gene-nutrient interaction, indicating that higher ω-3 consumption in individuals homozygous for the minor allele provides increased bioavailability of these products. Although the nutritional intervention was performed with ω-3 supplementation, it is emphasized that a greater consumption of food sources of this nutrient has a protective effect and may influence gene expression [30]. Thus, the pregnant women participating in this study were, by means of food and nutrition education, to increase their consumption of food sources of ω-3 and reduce excess dietary ω-6.

This guidance comes out of the theory of nutrigenomics, which is based on the interaction of dietary factors with the human genome, that is, the influence of nutrients on gene expression, thus allowing a better understanding of the mechanisms by which food components affect metabolic pathways and homeostatic control [7,8,31].

A recent cohort study on pregnant Spanish women evaluated the relationship between several polymorphisms in the FADS1 and FADS2 genes and gestational BMI, and whether these polymorphisms affected plasma LCPUFAs. The authors observed that women who had at least one minor allele of the FADS1 SNP had a higher risk of increased gestational BMI when compared to that of women homozygous for the major allele [12]. Despite the findings in the cohort study [12], the rs174575 and rs174561 SNPs did not reveal statistical significance, and the rs3834458 SNP (FADS2), which was adopted in the present study as a possible protective factor for women’s health, was not investigated in the cohort study on pregnant Spanish women [12].

In vitro studies are also employed to broaden knowledge about the interaction of dietary factors with the genome. Although they do not reproduce the reality that includes environmental and biological interactions such as occur in the multifaceted environment of human life, they are important means of knowledge production that contribute to the findings of population studies and promote understanding of the phenomenon.

It has been verified in vitro that the FADS1 and FADS2 genes have an influence on adipose tissue, body weight, and glucose homeostasis and that these are regulated by PUFAs [32]. Furthermore, adipocyte treatment with EPA and DHA can regulate the functionality of FADS genes, providing lower lipogenic gene activity and lipolytic gene expression [32,33], showing that ω-3 modulates the expression of FADS in human tissues, making their products (EPA and DHA) more bioavailable and revealing increased lipolytic activity that may have an influence on body weight [32,33].

In the present investigation, SNP rs3834458 (FADS2) showed opposite results on EPA concentrations when compared to SNPS rs174575 (FADS2) and rs174561 (FADS1). Although SNP rs3834458 (FADS2) revealed no association with women’s weight during pregnancy, it can indirectly be interpreted as a protective factor for this outcome, as it was related to a greater increase in products of PUFA ω-3 metabolism.

Other scientific investigations report results that corroborate those found in this study, revealing that the presence of the minor allele of this SNP provides higher concentrations of the product of ω-3 PUFAs [11], although this result is not consensual among the results of other investigations [34,35,36]. It is therefore necessary to expand the number and quality of investigations on these relationships in clinical and epidemiological settings.

In parallel with genetic influence, diet is another determining factor in the occurrence of nutritional problems. The current dietary characteristics of the Brazilian population reveal a risk in the balance of PUFAs. Among the pregnant women in this study, there was excessive consumption of processed foods and vegetable oil and reduced consumption of fruits and oilseeds. This consumption pattern is similar to that found in other populations of pregnant women, with processed and ultra-processed foods reported to be consumed more widely than others [17,37], a pattern that is responsible for a higher supply of ω-6, saturated fats, trans fats, and sodium and a low ω-3 content.

It is worth noting that this study is groundbreaking in Brazil, as it reveals an association between the rs174575 polymorphism (FADS2) and greater weight increase during pregnancy. This relationship may be justified by the influence of this polymorphism on gene transcription and consequently on the endogenous conversion of LC-PUFAs, reducing the bioavailability of the ω-3 series and its byproducts (EPA and DHA) and promoting the accumulation of the ω-6 series and its byproducts (ARA; ARA/LA).

In addition to genetic factors, the diet of pregnant women, characterized by excess consumption of food sources of ω-6, is a strong contributor to the expression of the results found. The genetic pattern, allied to environmental factors such as dietary profile, socio-demographic factors (higher maternal age), and unfavorable economic status (higher number of people in the home) during pregnancy, which are involved in the genesis of nutritional problems in this case, determines a higher probability of gestational overweight.

Based on the data in the analysis, pregnant women who were heterozygous and homozygous for the minor allele of the rs174575 polymorphism of FADS2 gained more weight during pregnancy and showed greater plasma concentrations of PUFA ω-6 (ARA and ARA/LA) and lesser ones of PUFA ω-3 (EPA; EPA/ALA; ALA). Pregnant women who were heterozygous and homozygous for the minor allele of the SNP rs3834458 (FADS2), on the other hand, showed larger concentrations of series ω-3 substrates, which indicates a protective factor for women’s health.

In this sense, nutrition initiatives are paramount in monitoring women in prenatal services in order to provide them with guidance on the role of adequate and healthy eating over the course of pregnancy. Special attention must be given to the consumption of nutritional sources of ω-3 fatty acids and to the reduction of consumption of processed and ultra-processed foods and consumption of vegetable oil and foods with higher ω-6 fatty acid content.

## Figures and Tables

**Figure 1 nutrients-14-01056-f001:**
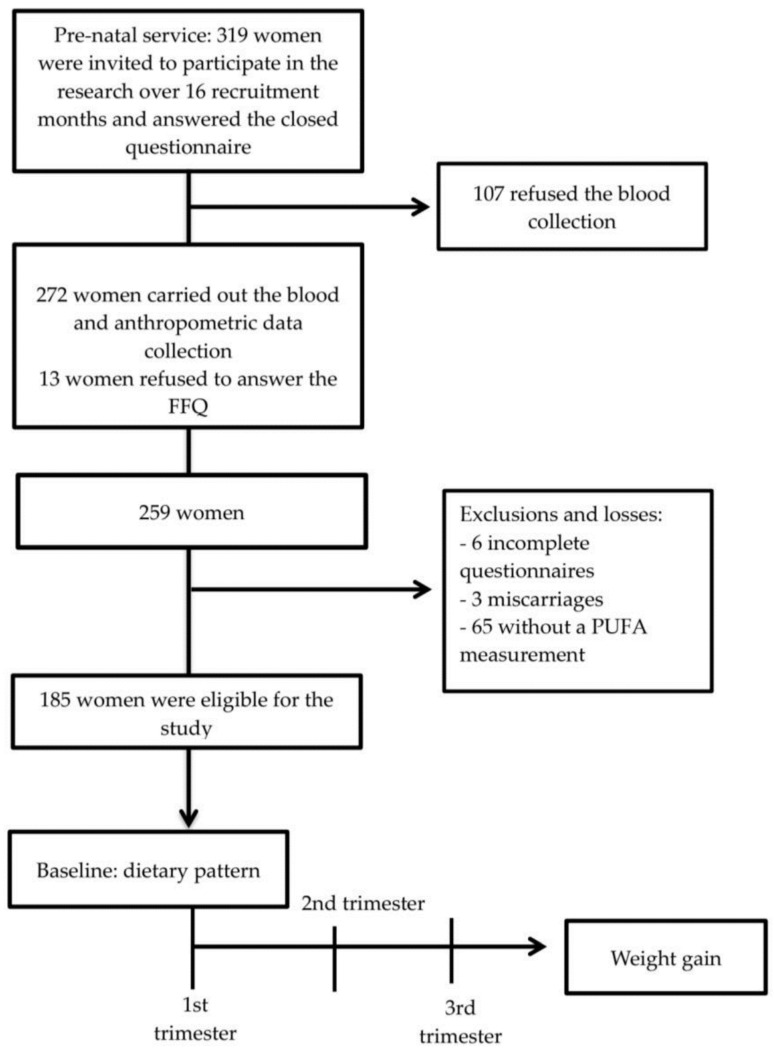
Flowchart of the design of the study conducted on pregnant women attending prenatal services in primary care units in Santo Antônio de Jesus, Bahia, Brazil, 2013–2015. FFQ-Food frequency questionnaire, PUFAs-polyunsaturated fatty acids.

**Table 1 nutrients-14-01056-t001:** Sociodemographic and nutritional characteristics according to weight gain during the gestational period, Brazil, 2013–2015.

Variables	No Excess Weight Gain		Excess Gestational Weight Gain		*p*-Value *
	Mean	SD	Mean	SD	
Age (years)	26.3	5.8	27.9	5.8	<0.001
Maternal education (years of study)	10.4	3.1	10.7	2.5	0.1317
Family income (minimum wage)	1302	803.4	1452.8	1334	0.0504
Number of residents	3.0	1.5	4.0	1.6	0.0020
Pre-gestational BMI	23.9	4.6	25.0	4.4	0.0025
Plasma ALA (%)	1.8	1.7	1.4	0.9	0.9958
Plasma LA (%)	13.7	5.8	12.8	4.8	0.9753
Plasma EPA (%)	2.5	1.9	2.2	1.8	0.9199
Plasma ARA (%)	1.7	1.3	1.5	1.2	0.9690
EPA/ALA	1.7	1.4	1.9	2.22	0.1443
ARA/LA	0.1	0.1	0.1	0.09	0.9837

ALA—α-linolenic acid. LA—linoleic acid. EPA—eicosapentaenoic acid. ARA—arachidonic acid. EPA/ALA—eicosapentaenoic acid and α-linolenic acid ratio. ARLA—arachidonic acid and linoleic acid ratio. BMI—body mass index. * Student’s *t* test for independent samples. *n* = 185:555 observations.

**Table 2 nutrients-14-01056-t002:** Structural equation modeling for association between FADS1 and FADS2 gene polymorphism plasma levels of polyunsaturated fatty acids and maternal weight throughout pregnancy, Brazil, 2013–2015.

Variables	Standardized Coefficients	*p* Value	95% CI
ALA←SNP rs174561	−0.25	0.039	−0.48; −0.013
ALA←SNP rs3834458	0.29	0.020	0.04; 0.52
ARA←SNP rs174575	0.24	<0.001	0.15; 0.33
ARA/LA←SNP rs174575	0.21	<0.001	0.12; 0.30
EPA←SNP rs174575	−0.12	0.009	−0.22; −0.03
EPA←SNP rs174561	−0.25	0.036	−0.480; −0.02
EPA←SNP rs3834458	0.27	0.026	0.10; 0.51
EPA/ALA←SNP rs174575	−0.12	0.014	−0.21; −0.10
IMCPG←SNP rs174575	0.17	<0.001	0.11; 0.26
Gestational weight←SNP rs174575	0.11	0.016	0.10; 0.19
Gestational weight←Age	0.28	<0.001	0.20; 0.35
Gestational weight←Alcohol consumer	0.14	<0.001	0.07; 0.22
Gestational weight←Mother’s education level	−0.10	0.012	−0.18; −0.02
Gestational weight←Trimester of pregnancy	0.12	0.002	0.04; 0.19
RMSEA: 0.0001			

LA—linolenic acid. EPA—eicosapentaenoic acid. DHA—docosahexaenoic acid. ARA—araquidonic acid. SNP—Single Nucleotide Polymorphism of the FADS1 and 2 genes. FADS—Polymorphisms of the desaturase genes. IMCPG—Pre-gestational body mass index, Kg/m^2^. RMSEA—root mean square error of approximation. *n* = 185:555 observation.

**Table 3 nutrients-14-01056-t003:** Direct, indirect, and total effects of FADS 1 and 2 SNPs mediated by PUFA plasma concentrations on the weight of pregnant women monitored by the NISAMI Cohort, Brazil, 2013–2015.

Variables	Direct Effect	Indirect Effect	Total Effect
Gestational weight←SNP rs174575	0.11	−0.01	0.09
Gestational weight←SNP rs 174561	−0.19	0.02	−0.17
Gestational weight←SNP rs 3834458	0.13	−0.01	0.11

SNP: Single Nucleotide Polymorphism; FADS: Desaturase gene polymorphisms; *n* = 185:555 observations.

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
