# Peer review of "FADS1 and FADS2 Gene Polymorphisms Modulate the Relationship of Omega-3 and Omega-6 Fatty Acid Plasma Concentrations in Gestational Weight Gain: A NISAMI Cohort Study"

_nutrients, 2022, doi:10.3390/nu14051056_

Round 1
Reviewer 1 Report
The authors attempt to describe the effect of fatty acid desaturase (FADS)1 and 2 on the association between omega-3 and omega-6 fatty acids plasma concentrations and gestational weight gain in a cohort of pregnant women in Santo Antonio de Jesus, Brazil. The positive features of the paper are the adequate sample size and the attention to details in collecting the anthropometric measurements. The lack of some rationale and details in the study design and lack of data on how both SNPs on FADS1 and 2 affect the outcome are the major drawbacks to the paper. The missing figure 2 might clarify some of the things described in the manuscript. In this paper, the authors assess the effect of each SNP separately. It is unclear how the presence of both SNPs for FADS1 and 2 affect the plasma concentrations of omega-3 and omega-6 and gestational weight gain. There are a few published reports in the literature that show some FADS1 and 2 are in linkage disequilibrium with one another. The authors should consider this possibility in their chosen SNPs and how it might affect the observed outcomes. In addition, the authors should consider including rationale on the why the choices of the three SNPs and including DHA in the analysis for omega-3 fatty acid. Overall, the manuscript offers an interesting perspective on the effect of genetic polymorphism of FADS1 and 2 separately impact plasma concentrations of eicosapentaenoic acid (EPA) and arachidonic acid (ARA) and gestational weight gain.
Major points:
- In Introduction section, the authors should consider including the importance of the SNPs chose in the study population or in general.
- In Materials and Methods section, p. 4, third paragraph, the authors should consider putting more details on how to quantify the fatty acids.
- In Materials and Methods, p. 5, subsection 2.8, it is unclear if the data are normally distributed.
- In Materials and Methods, p.6, subsection 2.8, Figure 2 is missing. The authors should consider including rationale on the choices of covariates in the model.
- In Discussion section, the major omega-3 fatty acids in our diet are EPA and DHA. The authors should include DHA in their analyses. If there is a particular reason in including only EPA, the rationale behind it is unclear.
Minor points
- The authors should define and abbreviate any words used throughout the manuscript when the words first appear in the text. For example, In Abstract section, the word “SNP” should be defined as single nucleotide polymorphism followed by “(SNP)” afterwards. The authors should use the abbreviations in the text after the definitions of the words.
- The authors should be consistent whether to put space or no space when referring to FADS1 or FADS2 throughout the text.
- In Abstract section, p.1, line 27, there is a mistype on the rs number.
- In Introduction section, p. 1, line 41, the authors should clarify what “This” refers to.
- In Materials and Methods section, p.2, line 81, the authors should use the plural word “samples”. In addition, there seems to be a mistype in “… 90% power (1-b) ...” It should be “… 90% power (1-b) …”
- In Materials and Methods section, p.3, figure 1, the title of figure 1 should be in bold fonts.
- In Materials and Methods section, p.3, line 96, the subtitle should be “Data collection”
- In Materials and Methods section, p.3, lines 97- 99, the sentence seems incomplete.
- In Materials and Methods section, p. 3, lines 101-103, the authors should consider rephrasing the sentence. The authors might want to say something like, “The time for blood collection in the clinical analysis laboratory in the city was then scheduled.”
- In Materials and Methods section, p.4, line 107, the word “after” should be omitted.
- In Materials and Methods section, p.4, lines 107 – 110, the authors should consider breaking the sentences. The sentences could be something like, “On the day before the blood collection, the pregnant woman received instruction on the exam protocol via telephone. The instruction included a 12-hour…”
- In Results section, p.6, lines 219-220, it is unclear the word “among the losses” refers to.
- Throughout the text, the authors should be consistent with the number of significant figures included in the data.
- In Results section, p.6, lines 222-223, the authors should consider presenting the data as mean standard deviation.
- In Results section, p.6, line 223, the letter “K” in kilogram should be a lower-case letter.
- In Results section, p.6, lines 225-228, the authors should consider rephrasing the sentence. The sentence could be something like, “The highest weight gain was observed in older pregnant women with highest pre-gestational BMI, who lived with the largest number of people at home.”
- In Results section, p.7, table 1, the authors should put the units of plasma levels of ALA, AT, EPA, ARA, and maternal education.
- In Results section, p. 8, lines 250-251, the use of word “meanings” in this sentence is very confusing. The authors should choose a different word.
- In Results section, p. 8, table 2, it is unclear what IMCPG means.
- In Results section, p. 9, line 263, there is a mistype of the value.
- In Results section, p.9, line 265, it is unclear in table 2 which parameter refers to the pre-pregnancy BMI.
- In Discussion section, p.9, line 287, the word “smaller” should be replaced with “lower”.
- In Discussion section, p.11, lines 336-337, the authors should consider rewording the sentence, “Thus, the pregnant women who composed the sample od this study were …” The sentence could be something like, “Thus, the pregnant women participated in this study were…”
- In Discussion section, p.11, lines 345-346, the space should be omitted.
- In Discussion section, p. 11, line 353, the word “in vitro” should be italicized.
- In Discussion section, p. 12, line 358, the word “in vitro” should be italicized.
Author Response
Author's Reply to the Review Report (Reviewer 1)
In Introduction section, the authors should consider including the importance of the SNPs chose in the study population or in general.
Comment: Added a paragraph on lines 64 to 66 of the introduction.
In Materials and Methods section, p. 4, third paragraph, the authors should consider putting more details on how to quantify the fatty acids.
Comment: Information on fatty acid dosages was added
In Materials and Methods, p. 5, subsection 2.8, it is unclear if the data are normally distributed.
Comment: Added a paragraph about data normality information.
In Materials and Methods, p.6, subsection 2.8, Figure 2 is missing. The authors should consider including rationale on the choices of covariates in the model.
Comment: Figure 2 was added to the text and the justification for the inclusion of the variables was based on the theoretical evaluation of the literature, as well as on the results of the bivariate analysis, which had a value of p < 0.2.
In Discussion section, the major omega-3 fatty acids in our diet are EPA and DHA. The authors should include DHA in their analyses. If there is a particular reason in including only EPA, the rationale behind it is unclear.
Comments: DHA was included in the bivariate analysis, however, it did not present criteria for entry in the multivariate analysis (p value <0.2), even so, due to theoretical strength, this variable was included, but did not show statistical relevance. In addition, the authors presented in the table only the results that were associated. This detail was added to the methodology.
In the discussion when fatty acids are mentioned, omega 3, DHA is not disregarded. We can cite points of the discussion that prove this statement:
Lines 304 to 307; 331 to 335
Minor points
The authors should define and abbreviate any words used throughout the manuscript when the words first appear in the text. For example, In Abstract section, the word “SNP” should be defined as single nucleotide polymorphism followed by “(SNP)” afterwards. The authors should use the abbreviations in the text after the definitions of the words.
Comment: Suggestion accepted
The authors should be consistent whether to put space or no space when referring to FADS1 or FADS2 throughout the text.
In Abstract section, p.1, line 27, there is a mistype on the rs number.
Comment: Suggestion accepted
In Introduction section, p. 1, line 41, the authors should clarify what “This” refers to.
Comment: The text was amended to make it clearer that we were referring to pre-gestational obesity and excess weight gain during pregnancy.
In Materials and Methods section, p.2, line 81, the authors should use the plural word “samples”. In addition, there seems to be a mistype in “… 90% power (1-b) ...” It should be “… 90% power (1-b) …”
Comment: Suggestion accepted
In Materials and Methods section, p.3, figure 1, the title of figure 1 should be in bold fonts.
Comment: Bold was added to the title of figure 1.
In Materials and Methods section, p.3, line 96, the subtitle should be “Data collection”
Comment: Suggestion accepted
In Materials and Methods section, p.3, lines 97- 99, the sentence seems incomplete.
Comment: Suggestion accepted
In Materials and Methods section, p. 3, lines 101-103, the authors should consider rephrasing the sentence. The authors might want to say something like, “The time for blood collection in the clinical analysis laboratory in the city was then scheduled.”
Comment: Suggestion accepted
In Materials and Methods section, p.4, line 107, the word “after” should be omitted.
Comment: Suggestion accepted
In Materials and Methods section, p.4, lines 107 – 110, the authors should consider breaking the sentences. The sentences could be something like, “On the day before the blood collection, the pregnant woman received instruction on the exam protocol via telephone. The instruction included a 12-hour…”
Comment: Suggestion accepted
In Results section, p.6, lines 219-220, it is unclear the word “among the losses” refers to. Throughout the text, the authors should be consistent with the number of significant figures included in the data.
Comment: The text has been changed to make it clearer. The loss that the authors refer to is the sample loss, women who were able to continue in the study, but dropped out for various reasons. The authors compared social, economic and outcome characteristics studied between the sample loss and the study sample. This methodological care guarantees greater reliability of the data.
In Results section, p.6, lines 222-223, the authors should consider presenting the data as mean standard deviation.
Comment: Suggestion accepted.
In Results section, p.6, line 223, the letter “K” in kilogram should be a lower-case letter.
Comment: Suggestion accepted.
In Results section, p.6, lines 225-228, the authors should consider rephrasing the sentence. The sentence could be something like, “The highest weight gain was observed in older pregnant women with highest pre-gestational BMI, who lived with the largest number of people at home.”
Comment: Suggestion accepted.
In Results section, p.7, table 1, the authors should put the units of plasma levels of ALA, AT, EPA, ARA, and maternal education.
In Results section, p. 8, lines 250-251, the use of word “meanings” in this sentence is very confusing. The authors should choose a different word.
Comment: Suggestion accepted.
In Results section, p. 8, table 2, it is unclear what IMCPG means.
Comentário: Significa Índice de massa corpórea pré-gestacional. Foi realizada alteração no texto.
In Results section, p. 9, line 263, there is a mistype of the value.
Comment: Suggestion accepted.
In Results section, p.9, line 265, it is unclear in table 2 which parameter refers to the pre-pregnancy BMI.
Comentário: Foi adicionado no rodapé da tabela o parâmetro é Kg/m2
In Discussion section, p.9, line 287, the word “smaller” should be replaced with “lower”.
Comment: Suggestion accepted.
In Discussion section, p.11, lines 336-337, the authors should consider rewording the sentence, “Thus, the pregnant women who composed the sample od this study were …” The sentence could be something like, “Thus, the pregnant women participated in this study were…”
Comment: Suggestion accepted.
In Discussion section, p.11, lines 345-346, the space should be omitted.
Comment: Suggestion accepted.
In Discussion section, p. 11, line 353, the word “in vitro” should be italicized.
Comment: Suggestion accepted.
In Discussion section, p. 12, line 358, the word “in vitro” should be italicized.
Comment: Suggestion accepted.

Reviewer 2 Report
FADS1 and FADS2 gene polymorphisms modulate the relationship of omega-3 and omega-6 fatty acid plasma concentrations in gestational weight gain: a cohort study
By Jerusa da Mota Santana et al.
Jerusa da Mota Santana and colleagues evaluated the effects of the single nucleotide polymorphisms (SNPs) in the genes encoding the fatty acid desaturase (FADS) 1 and 2 on gestational body weight gain of Brazilian mothers. They found that several SNPs found in Fads1 and Fads2 genes affect the concentrations of plasma PUFAs and that a SNP found in Fads2gene has positive effect on body weight gain through pregnancy. The findings might be important to the field; however, most of the results show replication of data in their previous paper (Carvalho et al., PLEFA, 2019), and I am not persuaded that the present data meet the high standards of Nutrients.
Comments:
- To dramatically improve their research significances, the authors would need to investigate the effects of dietary PUFAs, the SNPs of Fads1/2 genes, the concentrations of PUFAs including DHA in the plasma, and their combinations on gestational body weight gain.
- There are many typos in the manuscript. For example, "From this perspective, this study aims to evaluate the influence of single nucleotide polymorphisms of the FADS1 and FADS2 genes expressed in the plasma concentrations of PUFAs (ω-3 and ω-6) and the relationship with weight gain in pregnancy." ( 2, lines 65-67), and "GWG" would represent "gestational weight gain," not "excessive weight gain" (p. 1, line 37). The authors need to check whole the text carefully, and English usage of the text should be revised by a native English speaker.
Author Response
Author's Reply to the Review Report (Reviewer 2)
Jerusa da Mota Santana and colleagues evaluated the effects of the single nucleotide polymorphisms (SNPs) in the genes encoding the fatty acid desaturase (FADS) 1 and 2 on gestational body weight gain of Brazilian mothers. They found that several SNPs found in Fads1 and Fads2 genes affect the concentrations of plasma PUFAs and that a SNP found in Fads2gene has positive effect on body weight gain through pregnancy. The findings might be important to the field; however, most of the results show replication of data in their previous paper (Carvalho et al., PLEFA, 2019), and I am not persuaded that the present data meet the high standards of Nutrients.
Comment: The focus of this article is the influence of FADS1 and FADS2 gene polymorphisms on fatty acid concentrations and the impact on weight gain during pregnancy. A risk polymorphism associated with higher concentrations of w6 PUFA and greater weight gain during pregnancy was found. It is an innovative study for investigating a relationship that has been little studied in the area of women's health and that impacts health in later life cycles. Currently, the global epidemiological reality is of a high prevalence of obesity, especially in women, and the imbalance in weight gain during pregnancy has emerged as one of the main risk factors.
Comments:
To dramatically improve their research significances, the authors would need to investigate the effects of dietary PUFAs, the SNPs of Fads1/2 genes, the concentrations of PUFAs including DHA in the plasma, and their combinations on gestational body weight gain.
There are many typos in the manuscript. For example, "From this perspective, this study aims to evaluate the influence of single nucleotide polymorphisms of the FADS1 and FADS2 genes expressed in the plasma concentrations of PUFAs (ω-3 and ω-6) and the relationship with weight gain in pregnancy." ( 2, lines 65-67), and "GWG" would represent "gestational weight gain," not "excessive weight gain" (p. 1, line 37). The authors need to check whole the text carefully, and English usage of the text should be revised by a native English speaker.
Comment: English proofreading was carried out throughout the article

Round 2
Reviewer 1 Report
This reviewer would like to thank the authors to address the points presented in the previous report. There is a major concern related to the analysis of the samples. Polyunsaturated fatty acids are widely known to readily autoxidize with light and air. In the detailed description of the lipid extraction, the authors dried the samples in the oven overnight. In addition of the absence of antioxidant in the extraction, the exposure of samples to air for at least 8 hours is very concerning. Typically, the organic (chloroform) layer is dried under gentle stream of nitrogen and it takes very short time to dry out.
Author Response
Thanks for the contribution. Regarding the methodology for extracting the fatty acids, the samples were overnight in a culture oven with cut-off oxygen circulation, at 37°C. This information was promptly corrected in the text.
Reviewer 2 Report
The authors have addressed my previous concerns.
Author Response
Thanks for the contribution. Regarding the methodology for extracting the fatty acids, the samples were overnight in a culture oven with cut-off oxygen circulation, at 37oC. This information was promptly corrected in the text.
This manuscript is a resubmission of an earlier submission. The following is a list of the peer review reports and author responses from that submission.